## Research Article

# *Viperin* controls chikungunya virus–specific pathogenic T cell IFNγ Th1 stimulation in mice

Guillaume Carissimo[1] , Teck-Hui Teo[1], Yi-Hao Chan[1,2], Cheryl Yi-Pin Lee[1,2], Bernett Lee[1], Anthony Torres-Ruesta[1,3], Jeslin JL Tan[1], Tze-Kwang Chua[1], Siew-Wai Fong[1,4], Fok-Moon Lum[1], Lisa FP Ng[1,3,5]

**Chikungunya virus (CHIKV) has been a worldwide threat since its reemergence in La Reunion Island in 2004. Expression of the interferon-stimulated protein *Viperin* correlates with viral load burden in patients, and studies in mice have demonstrated its role to limit disease severity against CHIKV infection. Using *Viperin*[−/−] mice, we aimed to understand the contribution of *Viperin* to the T-cell immune response against CHIKV. CD4 T-cell depletion in *Viperin*[−/−] mice showed that increased late acute joint inflammation (5–8 d postinfection) was exclusively mediated by T cells. Specifically, CHIKV-infected *Viperin*[−/−] mice showed an increased INFγ Th1 profile of CD4 T cells, enhanced INFγ stimulation by APCs, an increased INFγ secretion profile in the joint microenvironment, and increased numbers of inflammatory monocytes in virus-infected joints compared with WT mice. Bone marrow grafting experiments showed that *Viperin* expression in both hematopoietic and non-hematopoietic cells is instrumental in reducing disease severity associated with a CD4 T-cell response.**

## Introduction

Chikungunya virus (CHIKV) is an alphavirus of the *Togaviridae* family that has become a worldwide public health issue since its reemergence in 2004 (Powers & Logue, 2007). Major outbreaks of CHIKV infection have spread across all islands in the Indian Ocean (Schuffenecker et al, 2006; Powers, 2011), India WHO, October 17, 2006; Ravi, 2006), countries in Southeast Asia (Hapuarachchi et al, 2010; Ng & Hapuarachchi, 2010; Pulmanausahakul et al, 2011), and more recently the Americas (Pan American Health Organization, 2015). Virus-infected patients typically present with a high fever, joint swelling that is associated with pro-inflammatory cytokine production and cellular infiltration during the acute infection phase (Ozden et al, 2007; Hoarau et al, 2010; Teng et al, 2015). Symptoms of arthralgia and myalgia can persist, in some cases, for up to several

years (Ozden et al, 2007; Hoarau et al, 2010; Teng et al, 2015). CHIKV viremia and the typical symptoms of the underlying pathology observed in infected patients can be recapitulated in mouse models following CHIKV infection via subcutaneous ventral footpad injection (Teo et al, 2013). Such CHIKV-infected mice show two peaks in joint footpad swelling, the first at 2–3 d postinfection (early acute) and the second at 5–8 d postinfection (late acute) that corresponds to the major swelling peak (Gardner et al, 2010; Morrison et al, 2011; Lum et al, 2013; Teo et al, 2013; Her et al, 2015). The early acute CHIKV-induced joint swelling is dependent on innate factors, such as *ISG15*, *IRF3*, *IRF7*, *TLR3*, and *Viperin* (Werneke et al, 2011; Schilte et al, 2012; Teng et al, 2012; Her et al, 2015), whereas late acute joint swelling is mediated by virus-specific CD4+ T cells (Teo et al, 2013). Regarding the latter, specific immunodominant pathogenic CD4 T-cell epitopes have been identified in the envelope E2 glycoprotein and the nonstructural protein nsP1 viral antigens (Teo et al, 2017).

Virus inhibitory protein, endoplasmic reticulum–associated, interferon-inducible (*Viperin*) is the product of the gene *RSAD2* (also known as *Cig5*) and is part of the interferon-stimulated gene (ISG) family (Helbig & Beard, 2014). *Viperin* is highly conserved and has antiviral functions in multiple organisms from fish to humans (Helbig & Beard, 2014). In humans, *Viperin* possesses antiviral activity against several clinically important viruses, including HIV-1, hepatitis C virus, and West Nile virus (Chin & Cresswell, 2001; Zhang et al, 2007; Szretter et al, 2011; Carlton-Smith & Elliott, 2012; Nasr et al, 2012; Tan et al, 2012; Teng et al, 2012; Wang et al, 2012; Helbig et al, 2013; Van der Hoek et al, 2017). More recently, *Viperin* was demonstrated to use a S-Adenosylmethionine-dependent mechanism to convert cytidine triphosphate to a nucleotide analog and function as a chain terminator of RNA polymerase of flaviviruses (Gizzi et al, 2018).

We have previously shown that *Viperin*[−/−] mice infected with CHIKV suffer more severe joint inflammation compared with infected WT controls (Teng et al, 2012). Both in vitro–infected primary tail fibroblasts and 1 dpi–infected joints of *Viperin*[−/−] mice express altered levels of various ISGs (Teng et al, 2012), compatible with an altered innate immune response to CHIKV. Although these actions

---

[1]Singapore Immunology Network, Agency for Science, Technology and Research, Singapore, Singapore   [2]National University of Singapore Graduate School for Integrative Sciences and Engineering, National University of Singapore, Singapore, Singapore   [3]Department of Biochemistry, Yong Loo Lin School of Medicine, National University of Singapore, Singapore, Singapore   [4]Department of Biological Science, Faculty of Science, National University of Singapore, Singapore, Singapore   [5]Institute of Infection and Global Health, University of Liverpool, Liverpool, UK

Correspondence: lisa_ng@immunol.a-star.edu.sg

of *Viperin* on innate immunity during initial CHIKV infection is known, the molecular mechanisms underlying enhanced joint inflammation during the late acute phase are unclear. In particular, little is known about the innate immune factors influencing the pathogenic CD4+ T-cell response that mediates the peak of joint swelling (Teo et al, 2013).

Here, the study aimed to investigate the role of *Viperin* in shaping the pathogenic CHIKV-specific CD4 T-cell adaptive immune response during late acute disease phase. Understanding this mechanism will help designing new therapeutic strategies that can reduce the pathogenic effect of CD4 T-cell responses during CHIKV infection.

## Results

### CD4 T cells mediate intensified joint swelling and reduce CHIKV-specific antibodies at 6 d postinfection (dpi) in *Viperin*$^{-/-}$ mice

To understand the contribution of CD4 T cells to joint swelling in CHIKV-infected *Viperin*$^{-/-}$ mice, CD4 T cells were depleted by intraperitoneal injection of a CD4-depleting antibody. CD4 T-cell depletion was confirmed by FACS (Fig S1A) before CHIKV inoculation via the joint footpad. Loss of CD4 T cells had a minimal impact on the first peak of joint swelling (2 dpi) but suppressed the second peak of joint swelling in *Viperin*$^{-/-}$ animals from 3 dpi onwards to

the levels of CD4 T cell–depleted WT animals (Fig 1A). Interestingly, injection of either isotype control IgG or CD4-specific IgG altered viremia that resulted in no detectable differences between WT and *Viperin*$^{-/-}$ animals (Fig S1B and E).

Titer and functionality of the CHIKV-specific antibody response is partly dependent on CD4 T cells (Lum et al, 2013). In addition, *Viperin*-deficient animals produced smaller amounts of IgG1 to ovalbumin stimulation but higher IgM and IgG amounts against envelope protein at 10 d post West Nile virus infection (Qiu et al, 2009; Szretter et al, 2011). Supporting these observations, we found that antibodies against CHIKV were produced in lower titers and were less neutralizing in CD4 T cell–depleted mice compared with nondepleted mice in both the WT and *Viperin*$^{-/-}$ groups (Fig 1B and C). In addition, a slight trend towards decreased IgG titers and neutralization capacity in *Viperin*$^{-/-}$ animals was observed in the non–CD4-depleted mice (Fig 1B and D). These findings suggest that consistent with previous observations (Qiu et al, 2009; Szretter et al, 2011), *Viperin* plays a minor role in antibody production.

### CD4 T cells infiltrate virus-infected joints in similar numbers

To understand the contribution of CD4 T cells in driving disease severity in *Viperin*$^{-/-}$ mice, immunophenotyping was performed to profile infiltrating leukocytes in the joints of CHIKV-infected mice at 6 dpi. No difference was detected in the total number of CD4 T cells in the joints of *Viperin*$^{-/-}$ mice compared with WT mice (Figs 2A and S2),

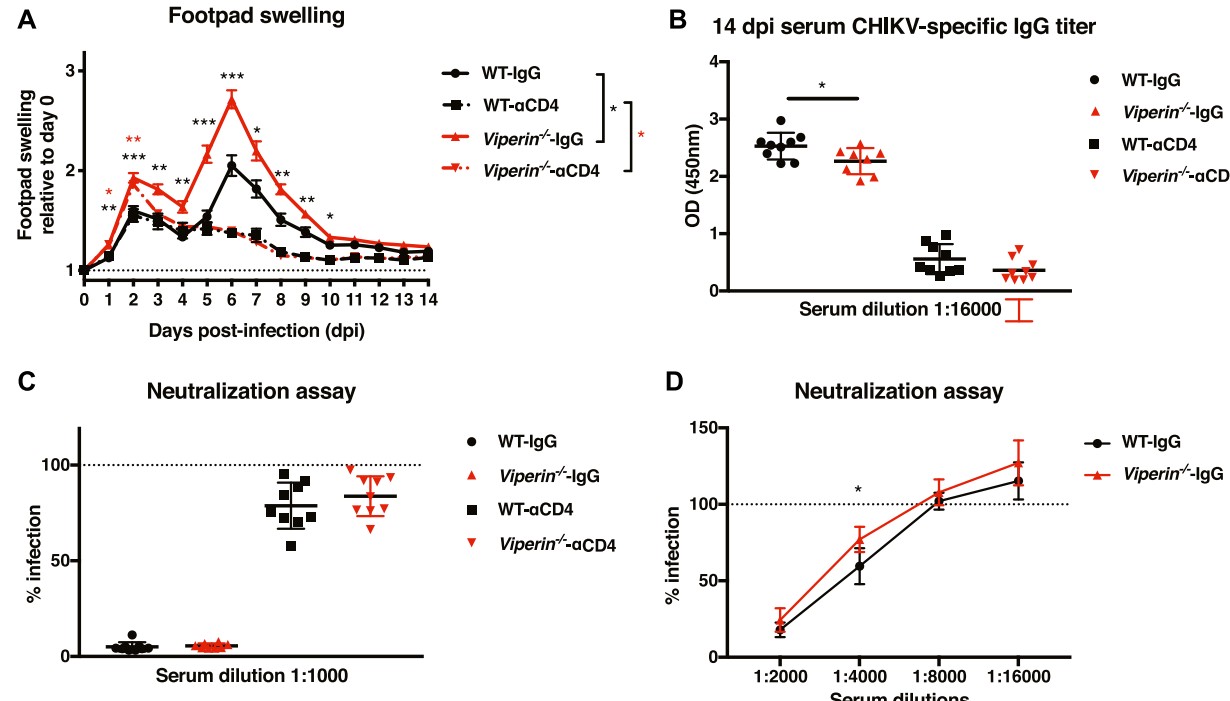

**Figure 1. CD4 T cells are responsible for the intensified joint pathology at 6 d postinfection.**
CD4 T cell–depleted or IgG control mice were infected with 1 × 10⁶ pfu CHIKV by ventral footpad inoculation at day 0 and monitored daily until 14 d postinfection (dpi). **(A)** Disease score of WT and *Viperin*$^{-/-}$ mice with (αCD4) or without (IgG) CD4 T-cell depletion calculated as the increase in footpad height × breadth relative to day 0. **(B)** CHIKV-specific IgG titers at day 14 dpi assessed by virion-based ELISA. **(C)** CHIKV neutralization capacity of individual serum samples at 14 dpi at a 1:1,000 dilution. **(D)** CHIKV neutralization capacity of serially diluted individual serums of non-CD4 T cell–depleted (IgG) mice. The data are representative of two independent experiments (n = 8–9 mice per group). Nonparametric Mann–Whitney *t* test was performed for each day between the groups indicated on the figure legend by the bar, and the significance is indicated on the figure by the * color (*P < 0.05, **P < 0.01, and ***P < 0.001).

as well as in the proportion of LFA-1+–activated CD4 T cells (Fig 2B). The numbers of total CD45+ leukocytes, CD8 T cells, neutrophils, and NK cells in the infected joints were also comparable between *Viperin*$^{-/-}$ and WT mice (Fig S3).

Interestingly, despite similar numbers of CD11b+Ly6C+ monocytes infiltrating the joints of CHIKV-infected mice (Fig 2C), the total number of CD11b+Ly6C+CD64+MHCII+ inflammatory monocytes significantly increased in the joints of *Viperin*$^{-/-}$ mice, both in absolute number per joint footpad (Fig 2D) and as a proportion of total infiltrating CD11b+Ly6C+ monocytes (Fig 2E). Together, these data show that despite similar cellular infiltration between the groups, the difference in pathology severity is likely due to a functional difference of the infiltrating T cells.

### *Viperin*$^{-/-}$ mice show an enhanced IFNγ Th1 response in the joints

As we did not detect differences in the numbers of CD4 T cells infiltrating the joints (Fig 2B), but CD4 T cells mediate CHIKV-induced foot swelling at 5–7 dpi (Gardner et al, 2010; Teo et al, 2013, 2015, 2017), we assessed if the infiltrating T cells were different between the

animals. Thus, we further phenotyped CD4 T cells infiltrating the joints in detail by assessing the proportion of CXCR3+Tbet+ (Th1 cells), CCR6+RoRγt+ (Th17), and CCR4+GATA3+ (Th2) in the CD3+CD4+CD44+ infiltrating cell population (Fig S4A). Interestingly, through this strategy, most CD4 T cells infiltrating the virus-infected joints at 6 dpi were CXCR3+Tbet+Th1 cells (Fig S4A). There were no detectable differences between WT and *Viperin*$^{-/-}$ mice (Fig 3A and B).

Therefore, we assessed the ability of these cells to secrete several cytokines (IFNγ, TNFα, IL-10, IL-4, and IL-17) upon nonspecific stimulation by PMA and ionomycin (Fig S4B). Upon stimulation, a higher proportion of IFNγ-producing T cells was detected in the joints of *Viperin*$^{-/-}$ animals (Fig 3C and D). In addition, a similar proportion of TNFα-producing cells was observed between the groups (Fig S5). We could not detect any IL-4–, IL-10–, or IL-17–producing cells in the virus-infected joints (Fig S4B), confirming that CD4 T cells infiltrating the joints during CHIKV infection are primarily Th1 cells.

### Similar tissue viral burden during peak joint swelling

It was hypothesized that a different viral burden in the joints could be responsible for the increased proportion of IFNγ-producing CD4

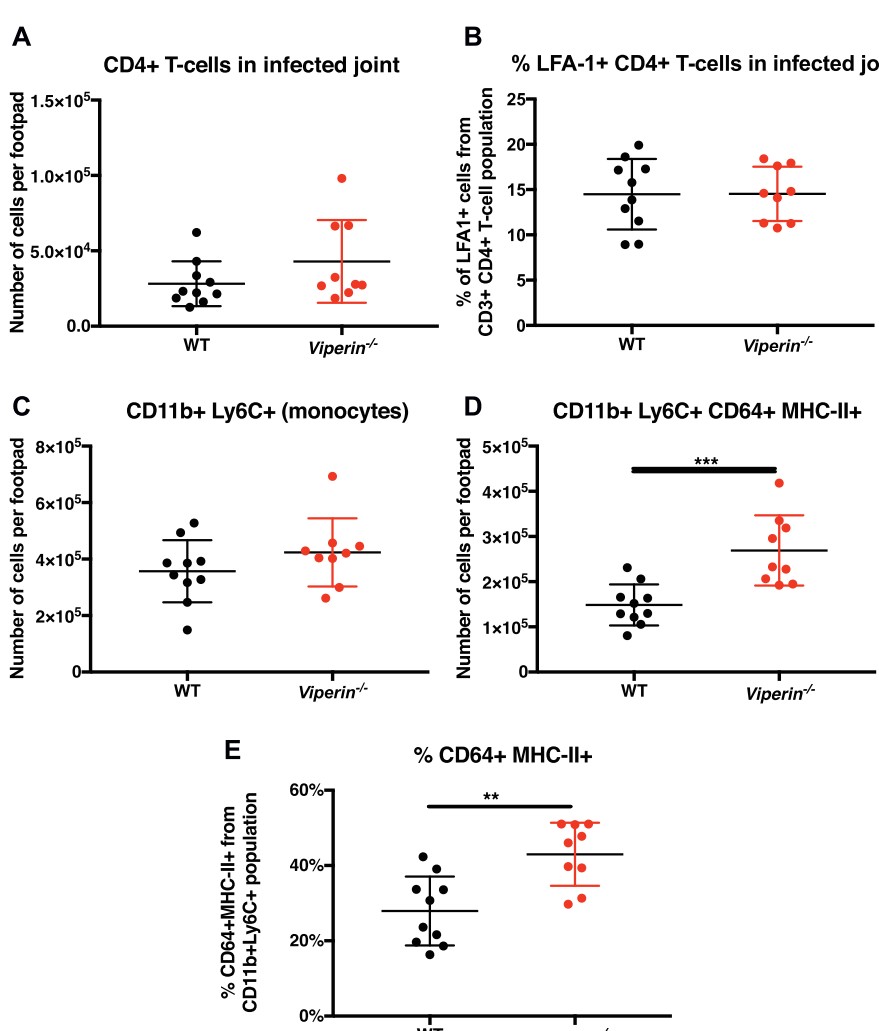

**Figure 2. CD4 T-cell infiltration of the joint is unaffected in *Viperin*$^{-/-}$ mice.**
Mice were infected with 1 × 10$^6$ pfu CHIKV by footpad injection and monitored daily until 6 dpi when the virus-infected footpad was harvested and analyzed by flow cytometry. **(A)** Number of CD11b–CD3+CD4+ T cells infiltrating the joint footpad at 6 dpi. **(B)** % LFA-1+ amongst CD4+ T cells infiltrating the joint footpad at 6 dpi. The data are representative of two independent experiments (n = 9–10 mice per group). **(C)** Number of infiltrating monocytes (CD11b+Ly6C+) per footpad. **(D)** Number of infiltrating CD11b+Ly6C+CD64+MHCII+ per footpad. **(E)** % of CD11b+ Ly6C+ CD64+ MHCII+ relative to CD11b+Ly6C+ per footpad. Gating strategy is presented in Fig S2, and the data were analyzed by Mann–Whitney nonparametric *t* test (**P < 0.01 and ***P < 0.001).

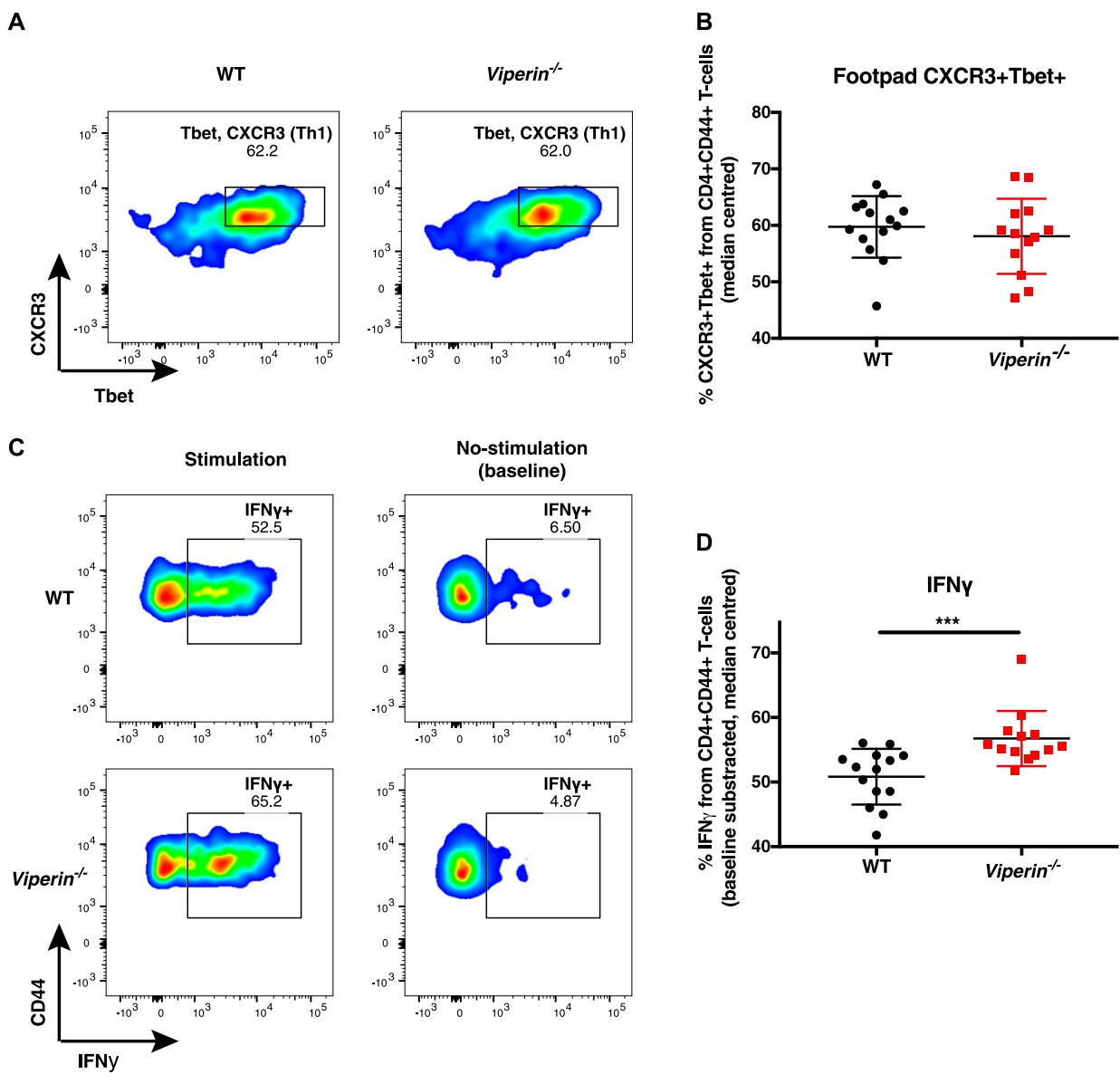

**Figure 3. A higher proportion of *Viperin*⁻/⁻ infiltrating Th1 cells are IFNγ producing.**
Mice were infected with 1 × 10⁶ pfu Fluc CHIKV by footpad injection and monitored daily until 6 dpi when the virus-infected joint footpad was harvested. **(A, B)** harvested cells were analyzed by flow cytometry. **(A)** Representative flow cytometry plot for CXCR3 and Tbet staining in WT and *Viperin*⁻/⁻ infiltrating CD44+ T cells, **(B)** median-centered proportion of CXCR3+Tbet+ across three independent experiments (n = 13–14 per group). **(C–D)** The cells were stimulated for 4 h with PMA and ionomycin before staining for common cytokines. **(C)** Representative flow cytometry plot for CD44 and IFNγ staining in WT and *Viperin*⁻/⁻ cells with and without stimulation. **(D)** Median-centered proportion of baseline-subtracted proportion of IFNγ+ T cells across three independent experiments (n = 13–14 per group). Gating strategies are presented in Fig S4, and the data were median-centered and analyzed by Mann–Whitney nonparametric *t* test (***$P < 0.001$).

T cells. To assess the tissue viral load in these animals, animals were infected with a *Firefly* luciferase–producing CHIKV infectious clone and monitored daily for luminescence in vivo (Fig S6A). Consistent with previous results (Teng et al, 2012), a higher tissue viral load was detected in the joints of *Viperin*⁻/⁻ mice at 1 dpi (Fig S6B), but no difference was seen at 6 dpi when the T cells infiltrate the joints (Fig S6D). Surprisingly, a lower tissue viral load was observed at 4 dpi in *Viperin*⁻/⁻ animals (Fig S6C). Together, these results show that the higher proportion of Th1-IFNγ–producing T cells is likely due to *Viperin* deficiency rather than local viral burden.

## *Viperin*⁻/⁻ APCs stimulate a higher amount of Th1-IFNγ cells

To identify the cause of the increased INFγ-Th1 response in *Viperin*⁻/⁻ animals, the proportion of CHIKV-specific IFNγ-producing CD4 T cells in the popliteal draining lymph node (pLN) of infected mice was assessed by ELISpot assays at 6 dpi. In this assay, virus-specific CD4 T cells secreting IFNγ after overnight stimulation with APCs and CHIKV can be detected. Importantly, the stimulation was performed in excess of APCs and viruses to minimize potential bias from infection. Consistent with the effect in the joints (Fig 3), a higher proportion of IFNγ-producing CD4 T cells was observed in

*Viperin*$^{-/-}$ mice compared with WT mice upon T-cell stimulation with WT APCs and CHIKV (Fig 4A). To understand whether this elevated IFNγ response was dependent on the APC genotype, ELISpot assay was performed on the same T cells using the two genotypes of APCs for stimulation. Interestingly, the number of IFNγ spots produced by WT T cells increased when stimulated with *Viperin*$^{-/-}$ APCs as compared with stimulation with WT APCs (Fig 4B).

The INFγ Th1 stimulatory capacity of APCs during CHIKV infection was then compared by calculating a stimulation ratio for each isolated T cell from individual mice as the number of spots induced by *Viperin*$^{-/-}$ APCs versus the number of spots induced by WT APCs for these particular T cells. A ratio value greater than one would directly correspond to a more efficient ability to stimulate IFNγ from Th1 T cells by *Viperin*$^{-/-}$ APCs. As expected, *Viperin*$^{-/-}$ APCs showed higher Th1 stimulation capacity as compared with WT APCs (Fig 4C). This effect was more pronounced for WT T cells (Fig 4C), likely reflecting their lower stimulation in vivo before isolation and restimulation. Together, these results show that Th1 cells undergo IFNγ-biased stimulation in *Viperin*$^{-/-}$ mice during the anti-CHIKV response, as a result of enhanced APC stimulation.

To validate that this IFNγ-biased stimulation was dependent on a virus–APC interaction, the ELISpot assay was then performed with the CHIKV-specific immunodominant peptide, E2EP3 (Teo et al, 2017) (Fig S7B and C). As expected, in this scenario without virus infection, the IFNγ stimulation ratio of either T-cell group was of 1 (Fig 4D).

These data functionally confirm that Th1-IFNγ–biased stimulation by *Viperin*$^{-/-}$ APCs (Fig 4C) is virus–APC dependent and not because of basal stimulation or antigen bias.

To verify whether this virus–APC interaction was inducing different soluble mediators levels in *Viperin*$^{-/-}$ condition, APCs were stimulated with CHIKV, and we quantified the cytokines released in the supernatant. As expected, CHIKV stimulation led to a higher production of pro-inflammatory cytokines such as IL-1β, IFNγ, IL-12p70, MCP-3, and ENA78 by *Viperin*$^{-/-}$ APCs (Fig S8). A modest increase of TNFα and IL-18 released by *Viperin*$^{-/-}$ APCs compared with WT APCs was also observed (Fig S8). Together, these results strongly suggest that the observed increased IFNγ T-cell stimulation is likely due to an increase of activating and polarizing soluble mediators released by the APCs during virus stimulation.

### Increased Th1-soluble mediators in *Viperin*$^{-/-}$ mice during infection

To verify that *Viperin*$^{-/-}$ mice generate an INFγ-Th1–stimulated microenvironment in vivo in the infected joints, immune mediators from footpad lysates were quantified at 6 dpi by multiplex bead-based array. We observed that the immune mediators IL-1α, IL-1β, IL-2, IL-4, IL-5, TNFα, M-CSF, LIF, IL-27, and IL-22 were all significantly increased in the joints of *Viperin*$^{-/-}$ mice (Fig 5), consistent with an increase of immune mediators in the joint footpad.

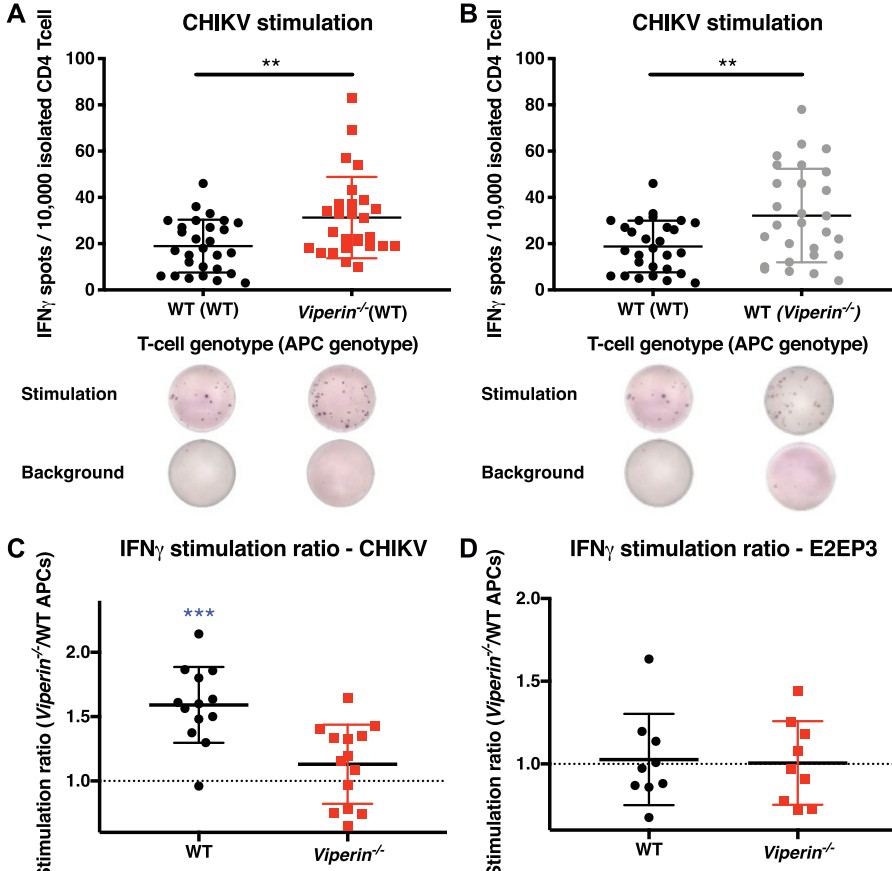

**Figure 4.** *Viperin*$^{-/-}$ **mice have an increased IFNγ Th1 response.**
Animals were infected with 1 × 10$^6$ pfu CHIKV by footpad injection and monitored daily until 6 dpi when the draining lymph nodes were harvested and CD4 T cells were isolated. ELISpot was performed on 10,000 T cells isolated from individual mice with two sets of APCs from naive WT mice and two sets of APCs from naive *Viperin*$^{-/-}$ mice stimulated with CHIKV, E2EP3 peptide, or no antigen control. The stimulation ratio was calculated for each mouse as a mean of IFNγ spots from *Viperin*$^{-/-}$ APC stimulation divided by the mean of IFNγ spots from WT deficient APC stimulation. **(A)** IFNγ spot count for CD4 T cells stimulated by WT APCs and CHIKV. A representative ELISpot image is shown. **(B)** IFNγ spot count for the same WT CD4 T cells stimulated by different APC genotypes and CHIKV. A representative ELISpot image for the same T cells is shown. **(C)** IFNγ stimulation ratio during CHIKV stimulation calculated by dividing the spots induced by *Viperin*$^{-/-}$ APCs by the spots induced by WT APCs for the same T-cell from each individual mouse (spot counts are presented in Figs 4A and S7A. (D) IFNγ stimulation ratio during E2EP3 peptide stimulation (spot counts are presented in Fig S7B and C). The results are representative of three independent experiments for CHIKV stimulation (n = 13–14 mice per group), two independent experiments for E2EP3 stimulation (n = 8 mice per group), and were analyzed by Mann–Whitney nonparametric t test (**P < 0.01), and stimulation ratio was analyzed by one-sample t test to be different from 1 (***P < 0.001).

The differences in systemic immune mediators leading to increased CHIKV pathology in *Viperin*$^{-/-}$ mice were also analyzed. Here, we quantified analytes in serum samples of *Viperin*$^{-/-}$ mice with or without CD4 depletion and the respective WT controls (Fig 6A). *Viperin*$^{-/-}$ and WT mice showed similar cytokine profiles during CHIKV infection. Consistent with an increase in INFγ-Th1 stimulation in *Viperin*$^{-/-}$ mice, a decrease in the serum levels of the Th2 cytokines IL-13 at 2 dpi, and IL-4 and IL-9 at 6 dpi was observed (Fig 6B). CD4 T-cell depletion had a marked impact on serum cytokine production with the expected decrease of IFNγ production and a general increase of pro-inflammatory cytokines at 2 dpi, 6 dpi, and 14 dpi. Up-regulated cytokines upon CD4 depletion differed between WT and *Viperin*$^{-/-}$, including IL-9, IL-18, and LIF, at 6 dpi for the knockout compared with WT (Fig 6A). Interestingly, basal levels of IL-18 and IL-22 were higher in *Viperin*$^{-/-}$ animals than WT animals before virus infection (Fig 6C), suggesting a potential predisposition of these mice for INFγ-Th1 stimulation. During infection, IL-18 levels remained consistently higher in *Viperin*$^{-/-}$ mice, whereas IL-22 levels increased in WT animals during infection to reach levels comparable with *Viperin*$^{-/-}$ animals at 6 dpi (Fig 6C). In addition, IL-22 had a particular profile in *Viperin*$^{-/-}$ CD4-depleted animals with very high levels at 2 and 6 dpi (Fig 6C). Together, these results confirm that *Viperin*$^{-/-}$ animals have an increased Th1 anti-CHIKV response systemically and in the infected microenvironment.

### *Viperin*$^{-/-}$ non-hematopoietic cells also control the intensity of CD4-mediated joint inflammation

Our findings thus far have shown that *Viperin*-deficient mice have an increased IFNγ Th1 anti-CHIKV response at the cellular and immune mediator levels (Figs 3, 4, 5, and 6). Furthermore, *Viperin*$^{-/-}$ APCs produced more inflammatory cytokines with increased Th1 stimulatory ability (Figs S7 and 4). We thus wanted to confirm if *Viperin* deficiency in the hematopoietic compartment alone is sufficient to elicit severe joint pathology during the peak of joint footpad swelling.

Chimeric mice between WT and *Viperin*$^{-/-}$ genotypes were generated by intravenously injecting freshly harvested bone marrow cells of each genotype in 6-wk-old irradiated mice from each genotype (Fig 7A). After verification of bone marrow reconstitution (Fig S9A and B), these mice were inoculated with CHIKV and monitored for joint footpad swelling and viremia for 14 d. WT animals engrafted with WT bone marrow (WT←WT) consistently presented with reduced joint footpad swelling compared with *Viperin*$^{-/-}$ mice engrafted with *Viperin*$^{-/-}$ bone marrow (*Viperin*$^{-/-}$← *Viperin*$^{-/-}$) (Fig 7B). WT animals with *Viperin*$^{-/-}$ bone marrow (WT←*Viperin*$^{-/-}$) and *Viperin*$^{-/-}$ animals with WT bone marrow (*Viperin*$^{-/-}$←WT) both showed an intermediate phenotype during the major joint swelling peak (Fig 7B). No effect was observed in viremia between any of the groups including full WT versus full *Viperin*$^{-/-}$, which could be explained by the age of these animals (Fig S9C). These results highlight the importance of *Viperin* in both hematopoietic and non-hematopoietic cells in shaping the pathogenic CD4 T-cell responses during the major peak of joint footpad swelling.

## Discussion

This study has confirmed that increased joint footpad pathology induced by CHIKV infection in *Viperin*$^{-/-}$ mice is mediated by increased CD4 T-cell polarization/stimulation towards Th1-IFNγ–producing cells. This T-cell polarization identified as IFNγ-producing Th1 (Fig 3), could be a result of the increased stimulation potential of *Viperin*$^{-/-}$ APCs via soluble mediators after viral stimulation (Figs 4 and 5, and S8). Consistent with this, it was recently shown in vitro that mice BMDM deficient for *Viperin* were more polarized to either M1 or M2 and had enhanced secretion of immune mediators upon stimulation (Eom et al, 2018). Similarly, *Viperin*$^{-/-}$ was shown to be necessary for inhibition of type I IFN production in macrophages (Hee & Cresswell, 2017). Consistent with an exacerbated polarization of *Viperin*$^{-/-}$ APCs, an increased pro-inflammatory environment was observed in the virus-infected joint (Fig 5) together with an increased secretion of soluble mediators by APCs during in vitro infection (Fig S8).

In addition, we observed an increased proportion of infiltrating monocytes differentiating into a CD64+MHCII+ inflammatory phenotype in the joints of *Viperin*$^{-/-}$ mice (Fig 2). This finding is consistent with our previous report showing increased F4/80 staining in histological sections of 6 dpi joints from *Viperin*$^{-/-}$ mice (Teng et al, 2012). During chronic *Leishmania major* infection, IFNγ secreted by infiltrating Th1 cells may be involved in the differentiation and/or function of inflammatory CD64+MHCII+ monocytes (De Trez et al, 2009). Supporting this hypothesis, CD4 T-cell depletion in CHIKV-infected mice reduced the numbers of this differentiated inflammatory CD64+MHCII+ monocyte population despite similar numbers of total CD11b+Ly6C+ monocytes (Lum FM et al, unpublished data). Therefore, a similar mechanism could explain the increase of this cell population observed in *Viperin*$^{-/-}$ pro-inflammatory microenvironment during CHIKV infection.

Interestingly, these monocyte-derived cells were shown to be important for normal CD4 Th1 activity in the lymph nodes (Nakano et al, 2009; Cheong et al, 2010), and important to activate/re-activate Th1 effector T cells in virus-infected tissues (Iijima et al, 2011). Thus, this study indicates that during CHIKV infection of *Viperin*$^{-/-}$ animals, the increased numbers of these cells in the joint footpad could be responsible for the observed CD4 T cell–mediated increased joint footpad swelling via increased restimulation in the infected tissue.

Importantly, we have previously shown that CD4 T cells did not induce the footpad pathology via an IFNγ pathway mechanism (Teo et al, 2013), which suggest a different mediator or effector mechanism of these cells in the pathology increase in *Viperin*$^{-/-}$ context. Interestingly, it was shown recently that granzyme A is a major promoter of T cell–mediated arthritic inflammation (Wilson et al, 2017). This makes granzyme A produced by these Th1-IFNγ–producing T cells that we identified an attractive effector candidate for the exacerbated joint pathology that we observed.

Surprisingly, we observed higher levels of known Th2 cytokines, IL-4 and IL-5, in *Viperin*$^{-/-}$ virus–infected joints (Fig 5), suggesting that the Th2 response should be increased in these animals. However, we could not detect either Th17 or Th2 cells infiltrating the virus-infected joints of either WT or *Viperin*$^{-/-}$ (Fig S4). It is possible

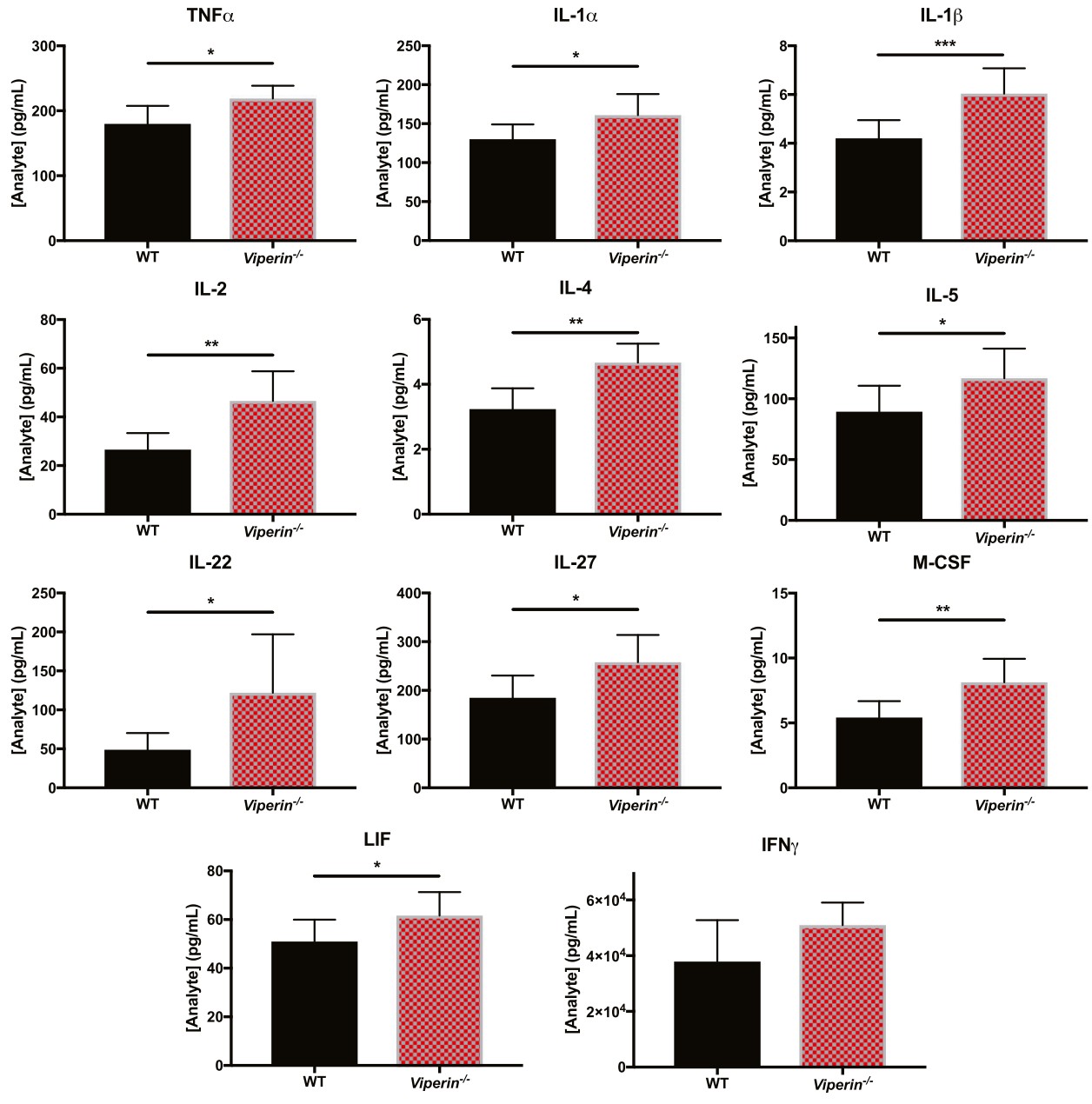

**Figure 5.  *Viperin*$^{-/-}$ joint footpads have higher levels of pro-inflammatory Th1 cytokines.**
Animals were infected with 1 × 10$^6$ pfu CHIKV by joint footpad inoculation and monitored daily until 6 dpi. Then, the virus-infected footpad was harvested and lysed in RIPA buffer with protease inhibitors followed by multiplex Luminex assay. The concentration of each analyte is expressed in pg/ml, where one footpad was lysed in 1 ml solution. The results are representative of two independent experiments (n = 7–8 mice per group) and analyzed by Mann–Whitney nonparametric one-tailed *t* test (**P* < 0.05, ***P* < 0.01, and ****P* < 0.001).

that this increased IL-4 and IL-5 in the microenvironment is a compensatory mechanism or an increased negative feedback loop to limit the increased IFNγ Th1 stimulation in *Viperin*$^{-/-}$ animals. Indeed, IL-4 levels in tissues were previously shown to reduce infiltration of Th1 cells (Lazarski et al, 2013). This would in turn explain why a similar infiltration of CD4 T cell numbers in WT and *Viperin*$^{-/-}$ mice was observed despite a higher IFNγ Th1 stimulation.

Upon CHIKV infection, *Viperin*$^{-/-}$ mice produced low serum levels of Th2 cytokines (IL-4, IL-5, and IL-9) and high levels of the Th1

cytokine IL-18. This Th1-biased serum cytokine phenotype corroborates a previous study where prolonged in vitro TCR stimulation with anti-CD3 and anti-CD28 resulted in lower Th2 cytokine secretion (IL-4, IL-5, and IL-13) by *Viperin*-deficient CD4 T cells compared with WT CD4 T cells (Qiu et al, 2009). Interestingly, in noninfected animals, the basal levels of IL-18 and IL-22 were found to be significantly higher in *Viperin*$^{-/-}$ compared with WT, which could indicate a basal inflammation or a predisposition to inflammation of these mice. In addition, IL-22 production was increased

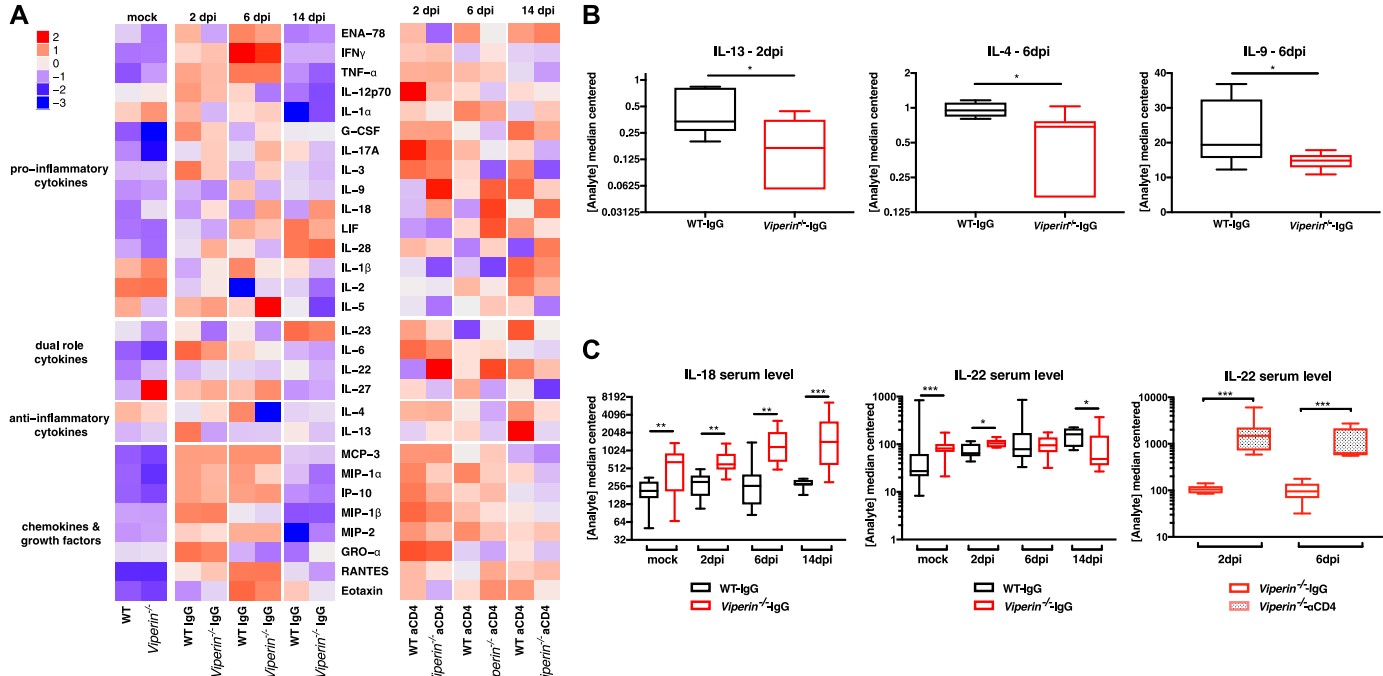

**Figure 6. *Viperin⁻/⁻* mice have high levels of pro-inflammatory cytokines but low levels of Th2 cytokines.**
Serum samples from individual mice used in CD4 T-cell depletion experiments (Fig 1) were analyzed by multiplex Luminex assay (36-plex). **(A)** Heat map of log-transformed analyte concentration values. **(B)** Median-centered serum concentrations of Th2 cytokines (pg/ml). **(C)** IL-18 and IL-22 median-centered serum levels (pg/ml). The results are representative of three independent experiments (n = 11–12 mice per group) and analyzed by Mann–Whitney nonparametric one-tailed *t* test for panels B and C (*$P < 0.05$, **$P < 0.01$, and ***$P < 0.001$).

following CHIKV infection only in CD4 T cell–depleted *Viperin⁻/⁻* mice, suggesting a potential involvement of *Viperin* in a negative signaling feedback loop on non–T-cell IL-22 production during CHIKV infection. Importantly, others have reported that *Viperin* can inhibit the macrophage interferon response (Hee & Cresswell, 2017), restrain BMDM polarization and cytokine production (Eom et al, 2018), modulate NF-κB and AP-1 expression in T cells (Qiu et al, 2009), and modulate TLR-7 and TLR-9 signaling in plasmacytoid dendritic cells (Jiang & Chen, 2011; Saitoh et al, 2011).

We also assessed the contribution of *Viperin* from non-hematopoietic cells in driving CHIKV disease pathology. By establishing a series of bone marrow grafts between WT and *Viperin⁻/⁻* mice, we functionally demonstrated that non-hematopoietic cells also participate in the increased pathogenic response of chimeric

mice (*Viperin⁻/⁻*←WT and WT←*Viperin⁻/⁻*), which experienced intermediate swelling compared with controls. These results suggest that *Viperin* expressed in non-hematopoietic cells also contributes to the intensity of the CD4 T cell–mediated virus-infected joint swelling. Non-hematopoietic cells have important roles in the innate immune anti-CHIKV defense such as TLR-3 and type I IFN expression (Schilte et al, 2010; Her et al, 2015). In light of evidence implicating non-hematopoietic cells in the control of T-cell memory, tolerance, and antigen-specific responses in the periphery (Carman & Martinelli, 2015; Humbert et al, 2016), it would be interesting to understand the mechanisms behind the control of the anti-CHIKV CD4 T-cell response by *Viperin* in non-hematopoietic cells.

Our results reinforce the notion of interplay between nonimmune and immune cells during CHIKV infection and highlight a pivotal role

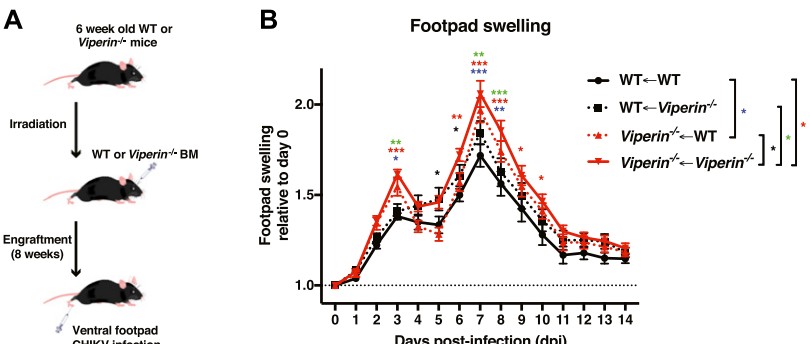

**Figure 7. Nonhematopoietic cells help regulate the pathogenic adaptive immune response during the second peak of joint swelling.**
WT and *Viperin⁻/⁻* chimeric mice were generated as indicated in the figure (genotype of the mice←genotype of the bone marrow). 8 wk after bone marrow graft, the mice were infected with 1 × 10⁶ pfu CHIKV by joint footpad inoculation. **(A)** Schematic representation of experimental protocol. **(B)** Disease score calculated as the increase of footpad height × breadth relative to day 0. The data are representative of two independent experiments (n = 10–12 mice per group) and analyzed by nonparametric Mann–Whitney *t* test, performed for each day between the groups indicated on the figure legend by the bar, and the significance is indicated on the figure by the * color (*$P < 0.05$, **$P < 0.01$, and ***$P < 0.001$).

for *Viperin* in influencing APC and T-cell immunity. Such control of IFNγ Th1 stimulation by *Viperin* during the anti-CHIKV response may open new avenues of research for other pathologies. For example, the immunopathogenicity of rheumatoid arthritis is driven by over-activated Th1 and/or Th17 T cells (Schulze-Koops & Kalden, 2001; Murphy et al, 2003). In that context, inhibition of T-cell costimulation (Linsley & Nadler, 2009) or disruption of Th17/T$_{REG}$ balance by blocking IL-6 pathway may be an effective treatment strategy (Schinnerling et al, 2017). Similarly, modulating endogenous *Viperin* activity to inhibit Th1 over-activation could be explored in rheumatoid/inflammatory arthritis or other disease models characterized by a pathogenic Th1 CD4 T-cell response. However, it remains to be determined in which specific cell type is *Viperin* expression responsible for the T-cell polarization phenotype.

Notably, *Viperin* enzymatic function was recently shown to be a convertase of CTP in 3′-deoxy-3′,4′-didehydro-CTP (ddhCTP) which was also reported to have chain reaction termination properties on RNA polymerase of flaviviruses (Gizzi et al, 2018). Our results, taken together with the observed phenotypes of *Viperin*$^{-/-}$ on macrophage polarization and cytokine production (Eom et al, 2018), interferon response signaling (Hee & Cresswell, 2017), and TLR-7/9 signaling pathway (Jiang & Chen, 2011; Saitoh et al, 2011), raise very intriguing questions on the main role of *Viperin* during immune responses. Is the enzymatic activity of *Viperin* linked to these immune regulatory functions, which would imply a role of CTP or ddhCTP in immune signaling and/or polarization, or are *Viperin* functions on immune responses unrelated to this enzymatic activity but rather to other *Viperin* protein partners because the SAM domain of *Viperin* is dispensable for flavivirus restriction (Helbig et al, 2011, 2013)? Deeper understanding of these mechanisms and their interplay during immune response will aid design strategies and treatments to target pathologies linked to over-activation of specific immune cells.

## Experimental procedures

### Mice

Female C57BL/6 WT and *Viperin* knockout (*Viperin*$^{-/-}$) mice aged 3–6 wk on a C57BL/6 background were used for all experiments. *Viperin*$^{-/-}$ mice were generated as described previously (Qiu et al, 2009). Briefly, heterozygous mutant mice with a neomycin cassette replacing exon 1 and 2 of *Viperin* were backcrossed in C57BL/6 mice seven times before establishment of homozygote line. All experimental procedures were approved by the Institutional Animal Care and Use Committee (IACUC, 151038) of the Agency for Science, Technology, and Research, Singapore, in accordance with the guidelines of the Agri-Food and Veterinary Authority and the National Advisory Committee for Laboratory Animal Research of Singapore.

### Viruses

The CHIKV isolate LR2006-OPY1 used here was originally isolated from a French patient returning from La Reunion Island during the 2006 outbreak (Bessaud et al, 2006). Virus was propagated in C6/36 cultures as previously described (Teo et al, 2013). Infectious clone of this isolate containing ZsGreen (Varghese et al, 2016; Lum et al, 2018) or *Firefly* luciferase (Fluc) (Pohjala et al, 2011; Teo et al, 2013) under the control of a subgenomic promoter was used for neutralization assays or mice infection, respectively. ZsGreen infectious clone was produced in African green monkey kidney epithelial cell clone E6 (Vero-E6, ATCC-CRL-1586) cells. Fluc virus was produced in C6/36 cultures as previously described (Teo et al, 2013). The virus titer of all viral stocks used was determined using standard plaque assays with Vero-E6 cells.

### Cell culture

Vero-E6 and human embryonic kidney clone 293T (HEK 293T, ATCC-CRL-3216) cells were cultured in DMEM supplemented with 10% FBS (Gibco). ELISpot assays were performed in RPMI (Gibco) supplemented with 10% FBS and 1% penicillin–streptomycin (Gibco). An *Aedes albopictus* mosquito cell line (C6/36, ATCC-CRL-1660) was cultured in Leibovitz's L-15 medium (Life Technologies) supplemented with 10% FBS. All cells were maintained at 37°C with 5% CO$_2$, except for the C6/36 cell line, which was maintained at 28°C without CO$_2$ supplementation.

### Animal studies

All animals were bred and housed under specific pathogen-free conditions at Biological Resource Center (A*STAR). Mice were inoculated subcutaneously in the ventral side of the right hind footpad towards the ankle, with 10$^6$ PFUs CHIKV (in 30 $\mu$l PBS). The viral load in tail blood samples was monitored daily from 1 dpi to 8 dpi, and then on alternate days until 14 dpi. Joint swelling of the footpad was scored daily from 0 to 14 dpi, as previously described (Teng et al, 2012; Teo et al, 2013; Her et al, 2015). Measurements were made for both the height (thickness) and the breadth of the foot and were quantified as (height × breadth). The degree of swelling was expressed as the relative increase in footpad size compared with preinfection (day 0), using the following formula: [(x − day 0)/day 0], where x is the quantified footpad measurement for each respective day.

Tissue replication was assessed by bioluminescence signals using an in vivo bioluminescence imaging system (IVIS Spectrum; Xenogen Corporation, Alameda, CA). Luciferase substrate, d-luciferin potassium salt (Caliper Life Sciences), was dissolved in PBS at a concentration of 5 mg/ml. Mice were anesthetized in an oxygen-rich induction chamber with 2% isoflurane. Measurements were performed 5 min after s.c. injection of 200 $\mu$l luciferin solution. Foot imaging was performed with the animal in a dorsal position with a field of view of 13.1 cm, open filter, and auto exposure settings. For bioluminescence quantification, regions of interest were drawn using the autodraw function with a threshold at 24% using the Living Image 4.5.4 software and the total radiance (p/s) was determined for each infected footpad.

CD4$^+$ T-cell depletion was performed as previously described (Teo et al, 2013). Briefly, each mouse was injected (i.p.) with 500 $\mu$g CD4-depleting antibody (InVivoPlus rat anti-mouse CD4, Bio X Cell, #BP0003-1) or rat IgG control (Sigma-Aldrich) on −1 and +4 dpi. CD4$^+$ T-cell depletion was assessed by flow cytometry before CHIKV inoculation (day 0) and at 10 dpi (Fig S1). Animals showing incomplete CD4 depletion at day 0 received a repeated dose of anti CD4-depleting antibody on the same day.

For bone marrow chimera studies, 6-wk-old recipient mice were irradiated twice with 600 Rad (4 h apart) and injected (i.v.)

with $4 \times 10^6$ donor bone marrow cells, as previously described (Sreeramkumar & Hidalgo, 2015). To test for successful adoptive bone marrow cell transfer, CD45.1 (WT) and CD45.2 ($Viperin^{-/-}$) staining on 10 $\mu$l tail blood was assessed by flow cytometry 7.5 wk post engraftment (Fig S9), virus infection was performed by ventral injection of $1 \times 10^6$ pfu at 8 wk post engraftment.

### Ethics statement

All animal procedures and experiments were reviewed and approved by the Institutional Animal Care and Use Committee (IACUC, 18353) in accordance with the guidelines of the Agri-Food and Veterinary Authority and the National Advisory Committee for Laboratory Animal Research of Singapore.

### Serum collection

Serum was collected from individual mice by retro-orbital bleeding at the times indicated. After clotting, two centrifugation steps were performed to collect the serum without cell contaminants. Aliquots were stored at –20°C until Luminex assays were performed, and other aliquots were heat-inactivated at 56°C for 30 min before ELISA and viral neutralization assays.

### Virion-based ELISA

Individual mouse serum antibody titers were assessed by standard virion-based ELISA, as previously described (Varghese et al, 2016; Lum et al, 2018). Briefly, CHIKV-coated ($10^6$ virions/well in 50 $\mu$l PBS) polystyrene 96-well MaxiSorp plates (Nunc) were blocked with PBS containing 0.05% Tween 20 (PBST) and 5% wt/vol nonfat milk for 1 h at 37°C. Mouse sera were serially diluted in antibody diluent (0.05% PBST + 2.5% wt/vol nonfat milk). Then, 100 $\mu$l diluted sera was added into each well and incubated for 1 h at 37°C, washed six times with PBST, and then incubated for 1 h at room temperature with 100 $\mu$l anti-mouse IgG HRP-conjugated antibody (Santa Cruz) before a final six washes in PBST. ELISA assays were then developed using TMB substrate (Sigma-Aldrich) and terminated with Stop Reagent (Sigma-Aldrich). Absorbance was measured at 450 nm using a TECAN Infinite M200 microplate reader (Tecan) and analyzed using Magellan software (Tecan).

### Neutralization assay

The neutralizing activity of antibodies from individual mouse sera was tested in triplicate and analyzed by immunofluorescence-based cell infection assay in HEK 293T cells, as previously described (Varghese et al, 2016; Lum et al, 2018). Briefly, HEK 293T cells were plated at a density of 30,000 cells per well of a 96-well plate. CHIKV isolate LR2006-OPY1 tagged with ZsGreen (multiplicity of infection [MOI] 5) was incubated with the individual mouse sera at the dilutions indicated, at 37°C for 1 h with gentle agitation (maximum 350 rpm). After incubation, the cell supernatant was removed and replaced with virus plus serum mixture for 16 h. Then, the cells were resuspended and washed in PBS before flow cytometric analysis. Data were acquired on the FITC channel using a MACSQuant Analyzer (Miltenyi Biotec), and the results were analyzed with FlowJo v10.1 software (FlowJo, LLC). The percentage neutralization was calculated according to the equation [% infection = (% infection from neutralization group/% infection from virus infection group) × 100].

### Viral RNA extraction and quantification

Blood (10 $\mu$l) was collected from the tail vein and diluted in 120 $\mu$l PBS with 10 $\mu$L citrate-phosphate-dextrose solution (Sigma-Aldrich). Viral RNA was extracted using a QIAmp Viral RNA Kit (QIAGEN), according to the manufacturer's protocol. CHIKV viral genome copies were quantified by Taqman RT-qPCR targeting viral RNA negative sense at the nsP1 region, as previously described (Kam et al, 2012; Teo et al, 2013).

### Determination of CHIKV-specific IFNγ-producing CD4+ T cells

CHIKV-specific CD4+ T cells in the draining pLN were analyzed by IFNγ ELISpot assay, as previously described (Teo et al, 2013) with slight modifications. Briefly, CD4+ T cells were isolated from pLN cells using a CD4+ T-cell Isolation Kit (Miltenyi Biotec). Naive splenocytes were harvested by mashing the spleen on a 40-nm cell strainer in RPMI 10% FBS (Gibco), followed by red blood cell lysis (R & D Systems, WL2000). Isolated T cells ($1 \times 10^4$) were stimulated with $2 \times 10^5$ naive splenocytes with a final concentration of 30 U/ml IL-2 for 15 h with CHIKV virions (MOI of 10) or 30 $\mu$g/ml E2EP3 peptide (Teo et al, 2017). The spleens of two naive WT and two naive $Viperin^{-/-}$ mice were used separately for each ELISpot experiment to stimulate CD4 T cells isolated from each individual pLN. For each individual isolated CD4 T-cell stimulation ratio calculation, the mean of the two $Viperin^{-/-}$ splenocyte stimulations was divided by the mean of the two WT splenocyte stimulations. For virus stimulation, the data were generated during three independent experiments (14 mice in total per group); for E2EP3, the data were generated during two independent experiments (nine mice in total per group).

### Splenocyte cytokine production quantification

Splenocytes of WT and $Viperin^{-/-}$ mice were collected in a similar manner and infected with the same MOI as for the ELISpot assay. However, $2 \times 10^6$ splenocytes were plated per well in six-well plates without addition of isolated CD4 T cells. Clarified supernatant was collected at 15 h postinfection and subjected to multiplex Luminex assay.

### Leukocyte profiling in the joints

Mouse footpads were harvested in RPMI with 10% FBS and 1% penicillin–streptomycin, supplemented with DNase I (50 mg/ml; Roche Applied Science), collagenase IV (20 mg/ml; Sigma-Aldrich), and dispase (2 U/ml; Invitrogen) and digested for 3 h at 37°C before passing through a cell strainer (40 $\mu$m pore size). Purification using Percoll solution (p1644-IL; Sigma-Aldrich) diluted in RPMI (35% vol/vol) was performed before red blood cell lysis (WL2000; R&D Systems), followed by live-dead staining (AquaBlue fixable; Thermo Fisher Scientific) for 10 min. The cells were then washed and blocked in staining media (PBS 1% rat serum, 1% mice serum [vol/vol]) for 20 min, and then stained via two steps for 20 min with the reagents indicated in Table S1, followed by flow cytometry acquisition on a LSR Fortessa 5 (BD Biosciences). Analysis was performed with FlowJo 10.1, and the live single-cell gating strategy is indicated in Fig S2.

### T-cell profiling in the joints

Footpad cells were isolated as described previously without red blood cell lysis step to preserve viability. Following isolation, the

cells were stained with AquaBlue fixable (Thermo Fisher Scientific) for 10 min. The cells were then washed and stained for surface markers in staining media (MACS buffer with 2% rat serum, 2% mice serum [vol/vol]; Sigma-Aldrich) for 30 min on ice, washed, and then stained with streptavidin for 10 min on ice. The cells were then fixed in fixation/permabilisation buffer (eBioscience 00-5123, 00-5223) for 90 min on ice, washed two times with permabilisation buffer (eBioscience 00-8333) and blocked for 15 min on ice in permabilisation buffer with 2% rat serum and 2% mice serum (vol/vol). Intracellular staining was then performed in the permabilisation buffer with mouse/rat serum for 30 min on ice. The cells were then washed two times and acquired on a LSRII 5 laser (BD Biosciences). Antibodies and reagents for this panel are listed in Table S2. Analysis was performed with FlowJo 10.5.

Alternatively, after isolation cells were incubated in 50 $\mu$l of IMDM 10% FBS 1% PS with or without 40 ng/ml PMA (Sigma-Aldrich) and 1 $\mu$g/ml Ionomycin (Sigma-Aldrich) at 37°C. After 1 h, 50 $\mu$l of 2× of Brefeldin A (BioLegend, 420601) and 2× of GolgiStop (BD cytofix/cytoperm 554715) in the previously described was added to the cells and incubated for 3 more hours at 37°C. At which point cells were collected, washed, and stained with AquaBlue fixable (Thermo Fisher Scientific) for 10 min. The cells were then washed and stained for surface markers in staining media (MACS buffer with 2% rat serum and 2% mice serum [vol/vol], Sigma-Aldrich) for 30 min on ice, washed, and then stained with streptavidin for 10 min on ice. The cells were then fixed in fixation/permabilisation solution (BD cytofix/cytoperm 554715) for 30 min on ice, washed two times with BD Perm/Wash buffer and intracellular staining was for 30 min on ice in BD Perm/Wash buffer with 2% rat serum and 2% mice serum (vol/vol). The cells were then washed two times and acquired on a LSRII 5 laser (BD Biosciences). Antibodies and reagents for this panel are listed in Table S2. Analysis was performed with FlowJo 10.5.

### Cytokine profiling in the joints

Mice were perfused with PBS (Gibco), and then the footpads were collected and homogenized in 1 ml RIPA buffer (50 mM Tris-HCl, pH 7.4; 1% NP-40; 0.25% Sodium deoxycholate; 150 mM NaCl; and 1 mM EDTA) with 1× protease inhibitors (Roche Holding AG) using a gentleMACS M Tube and a gentleMACS Dissociator (Miltenyi Biotec). The cell lysates were then sonicated at 70% intensity for 15 s on ice (Branson Ultrasonics Sonifier S-450), and the supernatants were collected for cytokine and chemokine quantification. The data are expressed as pg/ml in the footpad lysate.

### Multiplex microbead immunoassay for cytokine quantification

Cytokine and chemokine levels in mice sera, footpad lysates, and splenocyte supernatants were measured simultaneously using a multiplex microbead-based immunoassay, ProcartaPlex mouse Cytokine & Chemokine 36-plex Panel 1A (EPX360-26092-901) (Thermo Fisher Scientific). Briefly, a four-fold serial dilution of the standard mix was prepared. Antibody magnetic beads were then aliquoted into 96-well plates and incubated with standards and 25 $\mu$l serum. After an overnight incubation at 4°C, the plates were washed using a magnetic wash station (BioTek) according to manufacturer's instructions, followed by incubation with a detection antibody mix. The plates were incubated for a further 30 min, washed, and then incubated for 10 min in the presence of the provided streptavidin-PE. Data were acquired on a Luminex FlexMap 3D instrument (Millipore) and analyzed using Bio-plex Manager6.0 software (Bio-Rad Laboratories) based on standard curves plotted through a five-parameter logistic curve setting. The cytokines and chemokines assayed included FN γ, IL-12p70, IL-13, IL-1$\beta$, IL-2, IL-4, IL-5, IL-6, TNF$\alpha$, GM-CSF, IL-18, IL-10, IL-17A, IL-22, IL-23, IL-27, IL-9, GRO $\alpha$, IP-10, MCP-1, MCP-3, MIP-1 $\alpha$, MIP-1 $\beta$, MIP-2, RANTES, Eotaxin, IFN $\alpha$, IL-15/IL-15R, IL-28, IL-31, IL-1$\alpha$, IL-3, G-CSF, LIF, ENA-78/CXCL5, and M-CSF.

For sera, the Luminex data were normalized using median centering (median concentration for each analyte was adjusted to the global analyte median) on a per-analyte basis to remove any plate effects. The concentrations were then logarithmically transformed before further analysis to ensure normality. Two-way ANOVAs were used to detect analytes that differ between sample groups over time. Heat maps were generated using Euclidean distances. All statistical computations for the Luminex analyses were performed in R version 3.1.1. Raw data for Luminex experiments is given in Supporting Information S10.

### Statistical analysis

When necessary, median centering was applied to the data to reduce variation between experiments. This is mentioned on the figure where this was applied. Statistical analyses were performed using GraphPad Prism 7.0c and 7.0e (GraphPad Software), using unpaired nonparametric Mann–Whitney statistical test for analyses unless otherwise specified in the methods or figure legends. The disease score and viremia is plotted as the means + SEM; all other data are plotted as the means + SD unless specified in the methods section. $P$ values considered statistically significant are represented with * for $P < 0.05$, ** for $P < 0.01$, and *** for $P < 0.001$.

## Supplementary Information

## Acknowledgements

The authors would like to thank the Singapore Immunology Network (SIgN) Flow Cytometry core for assistance with cytometry analyzes, Esther Mok from SIgN immune monitoring group for support in Luminex assay, and the SIgN mouse core for support in animal breeding. Prof Andres Merits from University of Tartu constructed and provided the infectious clone of tagged viruses used in this project. The authors also thank Dr. Jessica Tamanini of Insight Editing London for editing the manuscript before submission. This project is funded by Agency for Science, Technology and Research (A*STAR) core grant awarded to LFP Ng. S-W Fong is supported by MOE Type-A Tier 3 grant (R-154-000-697-112). Y-H Chan and CY-P Lee are supported by an A*STAR graduate scholarship (AGS) (Singapore). A Torres-Ruesta is supported by an A*STAR Singapore International Graduate Award (SINGA) scholarship. The funders had no role in the study design, data collection and analysis, decision to publish, or preparation of the manuscript. Flow cytometry and multiplex soluble protein assay platforms are part of the SIgN Immunomonitoring platform and supported by a BMRC IAF 311006 grant and BMRC transition funds #H16/99/b0/011.

**Life Science Alliance**

## Author Contributions

G Carissimo: conceptualization, data curation, formal analysis, validation, investigation, visualization, methodology, project administration, writing—original draft, review, and editing.

T-H Teo: conceptualization, formal analysis, validation, investigation, visualization, writing—original draft, review, and editing.

Y-H Chan: investigation, writing—original draft, review, and editing.

CY-P Lee: validation, investigation, and writing—review and editing.

B Lee: data curation, software, formal analysis, and writing—review and editing.

A Torres-Ruesta: investigation and writing—review and editing.

J Tan: investigation and writing—review and editing.

T-K Chua: investigation and writing—review and editing.

S-W Fong: validation, investigation, and writing—review and editing.

F-M Lum: conceptualization, investigation, and writing—review and editing.

LFP Ng: conceptualization, supervision, funding acquisition, project administration, and writing—original draft, review, and editing.

## Conflict of Interest Statement

The authors declare that they have no conflict of interest.

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
