## [Reviewer comments · Life Science Alliance]

Life Science Alliance

Viperin controls chikungunya virus-specific pathogenic T-cell IFN γ Th1 stimulation in mice

Guillaume Carissimo, Teck-Hui Teo, Yi-Hao Chan, Cheryl Lee, Bennett Lee, Anthony Torres-Ruesta, Jeslin Tan, Tze-Kwang Chua, Siew-Wai Fong, Fok-Moon Lum, and Lisa Ng
DOI: 10.26508/lsa.201900298

Corresponding author(s): Lisa Ng, Singapore Immunology Network and Lisa Ng, Singapore Immunology Network

Review Timeline:	Submission Date:	2019-01-08
	Editorial Decision:	2019-01-09
	Revision Received:	2019-01-10
	Accepted:	2019-01-10

Scientific Editor: Andrea Leibfried

Transaction Report:

Please note that the manuscript was previously reviewed at another journal and the reports were taken into account in the decision-making process at Life Science Alliance. Since the original reviews are not subject to Life Science Alliance's transparent review process policy, the reports and author response cannot be published.

January 9, 2019

RE: Life Science Alliance Manuscript #LSA-2019-00298-T

Prof. Lisa F.P. Ng
Singapore Immunology Network
Laboratory of Microbial Immunity
8A Biomedical Grove
#04-06 Immunos Building
Singapore 138648
Singapore

Dear Dr. Ng,

Thank you for submitting your revised manuscript entitled "Viperin inhibits CHIKV-specific pathogenic CD4 T-cell response by controlling IFN γ Th1 polarization" to Life Science Alliance. Your manuscript was previously reviewed at another journal twice, and you provided us with the two sets of reports obtained.

The reviewers appreciated the revision performed, but found your conclusions not decisive enough. This is not a concern for publication in our view, and we would be happy to publish your paper in Life Science Alliance pending final revisions necessary to meet our formatting guidelines (see below). We think it would be also good to introduce minor text changes to tone-down the role of viperin in Th1 polarization slightly in light of the persistent concern of reviewer #2.

When revising your work, please:

- remove the legends from the figures and upload these as individual files
- make sure that all figures are of adequate resolution, the ppt-to-pdf conversion is not working well for figure S4 (very blurry), please upload this figure directly as a high resolution pdf
- please make a single excel file out of File S10A-C (adding A, B, and C as separate tabs)
- please link your ORCID iD to your profile in our submission system - you should have received an email with instructions on how to do so

A. FINAL FILES:

-- High-resolution figure, supplementary figure and video files uploaded as individual files: See our detailed guidelines for preparing your production-ready images, <http://life-science-alliance.org/authorguide>

B. MANUSCRIPT ORGANIZATION AND FORMATTING:

Full guidelines are available on our Instructions for Authors page, <http://life-science-alliance.org/authorguide>

Sincerely,

Andrea Leibfried, PhD
Executive Editor
Life Science Alliance
Meyerhofstr. 1
69117 Heidelberg, Germany
t +49 6221 8891 502

e.a.leibfried@life-science-alliance.org
www.life-science-alliance.org

January 10, 2019

RE: Life Science Alliance Manuscript #LSA-2019-00298-TR

Prof. Lisa F.P. Ng
Singapore Immunology Network
Laboratory of Microbial Immunity
8A Biomedical Grove
#04-06 Immunos Building
Singapore 138648
Singapore

Dear Dr. Ng,

Thank you for submitting your Research Article entitled "Viperin controls chikungunya virus-specific pathogenic T-cell IFN γ Th1 stimulation in mice". I appreciate the introduced changes and it is a pleasure to let you know that your manuscript is now accepted for publication in Life Science Alliance. Congratulations on this interesting work.

*****IMPORTANT:** If you will be unreachable at any time, please provide us with the email address of an alternate author. Failure to respond to routine queries may lead to unavoidable delays in publication.*******

DISTRIBUTION OF MATERIALS:

Again, congratulations on a very nice paper. I hope you found the review process to be constructive and are pleased with how the manuscript was handled editorially. We look forward to future exciting

submissions from your lab.

Sincerely,
